# "Alperujo" Compost Improves Nodulation and Symbiotic Nitrogen Fixation of Soybean Inoculated with *Bradyrhizobium diazoefficiens*

Germán Tortosa [1,*], Socorro Mesa [1], María J. Delgado [1] and Carol V. Amaya-Gómez [2,*]

1 Department of Soil and Plant Microbiology, Estación Experimental del Zaidín (EEZ-CSIC), Profesor Albareda, 1, 18008 Granada, Spain
2 Corporación Colombiana de Investigación Agropecuaria-Agrosavia, Centro de Investigación La Libertad, Villavicencio 500009, Colombia
* Correspondence: german.tortosa@eez.csic.es (G.T.); camaya@agrosavia.co (C.V.A.-G.)

**Abstract:** The utilization of compost to enhance plant productivity and symbiotic nitrogen fixation (SNF) has been recognized as an effective alternative to synthetic nitrogen fertilizers. This environmentally sustainable method is readily accessible to farmers. This study investigated the effect of olive pomace ("alperujo", AL) compost on the nodulation and SNF of soybeans (*Glycine max* L.) and their natural symbiont (*Bradyrhizobium diazoefficiens*). For that, soybean plants were subjected to several doses of AL compost under controlled greenhouse conditions. At the end of the experiment, the dry weight of plant biomass (aerial part and roots), the number and fresh weight of nodules, and nitrogen and leghaemoglobin contents were analyzed. The application of AL compost significantly improved soybean growth, as demonstrated by an increase in both plant biomass and height. Furthermore, nodular leghaemoglobin content and nitrogen content were found to be enhanced by the addition of AL compost (7 and 40%, respectively), indicating an increase in nodule effectiveness and symbiotic efficiency. Our results provide clear evidence of the synergetic effect of AL compost on the soybean-*B. diazoefficiens* association, probably due to AL-compost improved soybean roots development, and rhizospheric organic matter and nutrients assimilation by rhizobia.

**Keywords:** compost; leghaemoglobin; legume; nitrogen fixation; nodules; rhizobia; symbiotic effectiveness

## 1. Introduction

Legumes are the main source of vegetable proteins for animal and human food and are also one of the main strategies for the sustainability of agricultural ecosystems [1]. An example is soybean (*Glycine max*), which accounts for about 50% of the total cultivated area and 68% of the world production of legumes [2]. On the other hand, legumes establish a mutual symbiosis with certain soil microorganisms known as rhizobia. The rhizobia-legume interaction encompasses a tight regulation process that only allows specific rhizobia strains to induce the formation of nodules in certain legumes. Inside the nodules, rhizobia carry out symbiotic nitrogen fixation (SNF), a process where the nitrogenase enzyme reduces atmospheric nitrogen ($N_2$) to ammonia ($NH_4^+$). This way, non-bioavailable nitrogen is converted into nitrogen compounds that plants can use for their growth and development [3].

Composting is a biotechnology that transforms organic waste into humic materials that can be utilized as fertilizers in agriculture. An example can be found in the two-phase olive pomace or "alperujo" (AL), which comes from the production of the olive oil and it is one of the most important organic wastes produced in Spain. It has been recently demonstrated that composting is a viable technology for its treatment and also as a feasible strategy for the production of solid and liquid organic commercial fertilizers [4,5].

In addition to rhizobia, other microbial strains that facilitate the acquisition of nutrients or produce phytohormones or compounds with antimicrobial activity are present [6,7]. Given the enhancement of nutrients and the mitigation of environmental stresses, compost improves the formation of legume nodules and, therefore, SNF. For instance, compost application to soils led to an increase in the number of nodules compared to soybeans grown without compost [8–10]. Considering all these benefits, compost has become a viable technology for the nutrient enrichment of soils, and it is also considered a feasible strategy to produce solid and liquid organic commercial fertilizers [4,5].

Compost is a very important source of organic matter, humic substances and plant nutrients, as well as a reservoir of beneficial microorganisms that can promote plant growth. Thus, the main objective of this work was to study the feasibility of AL compost to improve the SNF process and its efficiency in legume crops, specifically in soybean plants inoculated with *Bradyrhizobium diazoefficiens*, its natural endosymbiont.

## 2. Materials and Methods

### 2.1. AL Compost

The compost used in this study was previously prepared by mixing an equivalent amount of AL and sheep manure, as described in earlier work [4]. In short, a trapezoidal pile of 10 t was made and handled with a backhoe loader. A total of 7 turning operations were performed, which were more frequent during the bioxidative phase, and more time wise as the composting progressed. The pile humidity was maintained at values close to 40% thanks to a sprinkler irrigation system. The process lasted for a total of 22 weeks. The obtained compost had a slightly alkaline pH ($7.67 \pm 0.23$) and low salinity (EC of $1.11 \pm 0.08$ dS m$^{-1}$). Its organic matter content was $54.5 \pm 1.8\%$, with a significant lignocellulosic fraction and a remarkable humic content, with $78.30 \pm 3.39\%$ of humic acids. The total nitrogen content was $1.50 \pm 0.30\%$, essentially organic due to its low mineral content (ammonium, nitrate and nitrite). The AL compost did not show any phytotoxic properties, showing values of $93 \pm 1\%$ in the germination index (Zucconi test).

### 2.2. Cultivation of B. diazoefficiens

In this experiment, the parental strain (USDA 110) of *B. diazoefficiens* (U.S. Department of Agriculture, Beltsville, MD, USA) was used as soybean endosymbiont. This bacterium was routinely grown on PSY (peptone-salts-yeast) medium at 28 °C supplemented with chloramphenicol (15 µg L$^{-1}$) and arabinose (0.1% *w/v*), as previously described [11]. The preparation of the inoculum was made by growing *B. diazoefficiens* to stationary phase and then centrifuging it at $12,000 \times g$ at 4 °C for 10 min. Finally, the pellet was resuspended in sterile saline (NaCl, 0.9%) and achieved an optical density (600 nm) of 0.8, equivalent to about $10^8$ cells per mL.

### 2.3. Soybean Inoculation and Growth

Soybean (*Glycine max* L. Merr., cv. Williams) plants were inoculated with *B. diazoefficiens* USDA 110 as previously described [12]. Seeds were surface sterilized with ethanol (96%, *v/v*) and H$_2$O$_2$ (30%, *v/v*) and germinated in Petri dishes (1% agar, *w/v*) for 3 days at 28 °C. Seeds were transplanted into sand-filled pots (0.25 L) and inoculated with 1 mL of *B. diazoefficiens* inoculum. Plants were periodically watered (twice a week) with 40 mL of a nitrogen-free nutrient solution and grown under controlled light and temperature conditions in the Greenhouse and Growth Chamber Service of the Estación Experimental del Zaidín (EEZ-CSIC; https://www.eez.csic.es/en/green-houses; accessed on 15 May 2020) for 28 days [12] until the vegetative stage of V3-V4 (Figure 1).

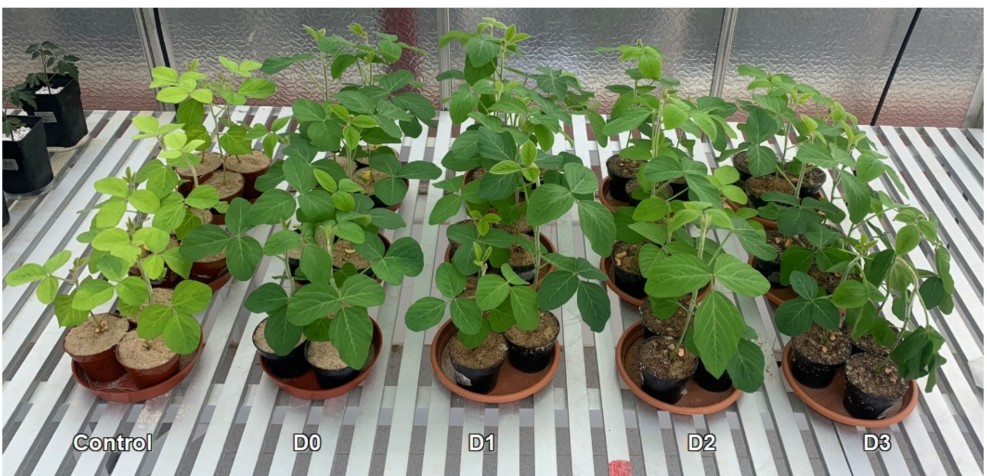

**Figure 1.** Soybeans before harvesting. Plants were grown with different doses of AL compost (D0, D1, D2 and D3). Non-inoculated plants grown in the absence of compost were used as the control.

### 2.4. Physiological Determinations

Shoot dry weight (SDW), root dry weight (RDW), nodule number (NN) and nodule fresh weight (NDW) were measured after harvesting (Figure 2). Nodule functionality and fitness were evaluated by fluorimetrically analyzing leghaemoglobin concentration (Lb) after the acidic reaction of the nodular fraction, as it is described in [12] (Figure 2). Furthermore, the nitrogen content in dry shoots ($N_{shoots}$), roots ($N_{roots}$), nodules ($N_{nodules}$) and seeds ($N_{seeds}$) were determined by the Dumas method using a LECO TruSpec CN Elemental Analyzer (USA) available at the Elemental Analysis Service of the EEZ-CSIC [11]. Total nitrogen accumulated ($T_{NA}$) was estimated according to the following formula: $T_{NA} = (N_{shoots} + N_{roots} + N_{nodules}) - N_{seeds}$. Data were expressed as mg N plant$^{-1}$ (dry weight).

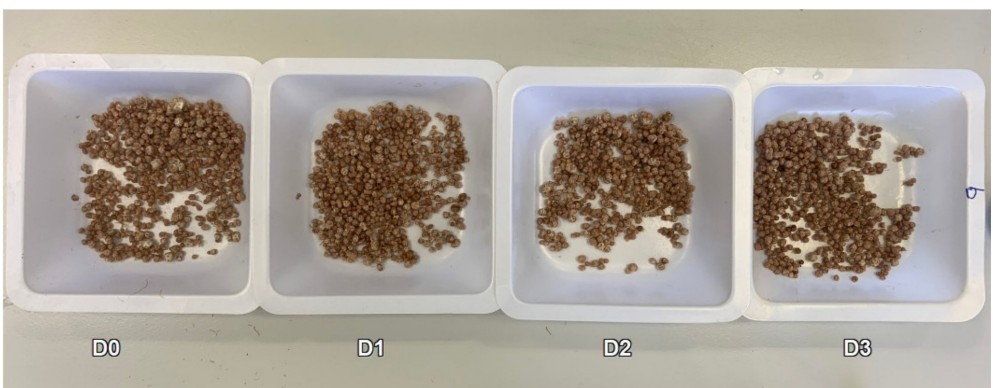

**Figure 2.** Soybean nodules after harvesting. Inoculated soybean plants were grown with different doses of AL compost: D0, D1, D2 and D3.

### 2.5. Experimental Design and Statistical Analysis

Five treatments were tested according to AL compost doses applied: Non-inoculated soybeans with no compost added (Control), inoculated soybean with *B. diazoefficiens* without compost (D0), inoculated soybean with *B. diazoefficiens* with compost (2.2 g per pot or 28 t ha$^{-1}$) (D1), inoculated soybean with *B. diazoefficiens* with compost (4.4 g per pot or 56 t ha$^{-1}$) (D2) and inoculated soybean with *B. diazoefficiens* with compost (6.6 g per pot or 84 t ha$^{-1}$) (D3).

The number of replicates per treatment was 9 and the assay was performed three times during one year of experimentation. Statistical treatment of the data consisted of the calculation of some descriptive statistics such as arithmetic mean and standard deviation,

as well as an inferential calculation based on the one-way analysis of variance (ANOVA) and Tukey tests ($p < 0.05$).

## 3. Results

The effect of AL compost on the growth of soybean plants was positive for all the parameters analyzed, especially plant biomass (Table 1). Both SDW and RDW increased by about 5 and 45%, respectively, compared to the D0 treatment. Improvements in plant height and trifoliate leaves were also observed, but without being statistically significant. Furthermore, the shoot to root ratio (SDW/RDW) decreased, which meant a differentiation in plant development produced by the different doses of AL compost added during growing.

**Table 1.** Effect of AL compost doses (D0, D1, D2 and D3) on plant height, trifoliate leaves, shoot (SDW) and root dry weight (RDW) and shoot to root ratio (SDW/RDW) of soybean plants. A set of non-inoculated plants (Control) grown without compost was included in the assay.

| Treatment | Plant Height (cm) | Trifoliate Leaves | SDW (g Plant$^{-1}$) | RDW (g Plant$^{-1}$) | SDW/RDW (Plant$^{-1}$) |
|---|---|---|---|---|---|
| Control | 15 b | 2 b | 1.23 c | 0.32 c | 3.84 a |
| D0 | 24 a | 3 a | 1.42 b | 0.45 b | 3.16 b |
| D1 | 28 a | 4 a | 1.51 a | 0.61 a | 2.48 c |
| D2 | 27 a | 4 a | 1.49 a | 0.67 a | 2.22 c |
| D3 | 26 a | 4 a | 1.48 a | 0.62 a | 2.39 c |

Values in a column followed by the same letter are not statistically different according to the Tukey test ($p < 0.05$).

Regarding soybean-*B. diazoefficiens* symbiosis, the addition of AL compost significantly improved NN by 7 to 20% compared to the D0 treatment (Table 2). In addition, a slight increase in nodular biomass was observed, especially in D2, which was 0.66 g NFW per plant. This data represented 8% more than the value obtained in D0. In terms of symbiotic efficiency, which was measured as Lb content, it was observed that AL compost produced a significant improvement depending on the dose applied. Lb concentration boosted from 6.82 in D0 to 7.12, 7.25 and 7.32 mg NFW$^{-1}$ in D1, D2 and D3 treatments, respectively.

**Table 2.** Effect of AL compost doses (D0, D1, D2 and D3) on nodule number (NN), nodules fresh weight (NFW), fresh weight per nodule (NFW/NN) and nodular leghaemoglobin content (Lb) of soybean plants. Non-inoculated plants grown without compost were used as a reference for the comparisons (Control).

| Treatment | NN (Plant$^{-1}$) | NFW (g Plant$^{-1}$) | NFW/NN (mg Nodule$^{-1}$) | Lb (mg NFW$^{-1}$) |
|---|---|---|---|---|
| Control | - | - | - | - |
| D0 | 65 b | 0.61 b | 6.82 c | 6.82 c |
| D1 | 70 ab | 0.63 b | 7.12 b | 7.12 b |
| D2 | 76 a | 0.66 a | 7.25 a | 7.25 a |
| D3 | 78 a | 0.62 b | 7.32 a | 7.32 a |

Values in a column followed by the same letter are not statistically different according to the Tukey test ($p < 0.05$).

Finally, the nitrogen accumulation in soybean biomass was also determined. AL compost addition significantly increased nitrogen content in shoots, roots and also nodules of soybean plants inoculated with *B. diazoefficiens* (Table 3). $N_{shoot}$ rose 1.2–1.3 fold with AL compost compared to D0 treatment. Similarly, $N_{root}$ was around 1.4 and 1.6 fold, and $N_{nodules}$ was close to 1.3 and 1.5 fold. Particularly, D2 was the best of the AL compost treatments for N acquisition in soybean plants, with $38.60 \pm 2.1$ mg plant$^{-1}$ of $T_{NA}$ (Figure 3).

**Table 3.** Effect of AL compost doses (D0, D1, D2 and D3) on nitrogen content in soybean shoots ($N_{shoot}$), roots ($N_{roots}$) and nodules ($N_{nodules}$).

| Treatment | $N_{shoot}$ (mg Plant$^{-1}$) | $N_{root}$ (mg Plant$^{-1}$) | $N_{nodules}$ (mg Plant$^{-1}$) |
|---|---|---|---|
| D0 | 21.47 b | 4.31 c | 2.31 c |
| D1 | 26.21 a | 5.89 b | 3.12 b |
| D2 | 28.26 a | 6.78 a | 3.56 a |
| D3 | 26.31 a | 6.04 ab | 2.91 b |

Values in a column followed by the same letter are not statistically different according to Tukey test ($p < 0.05$).

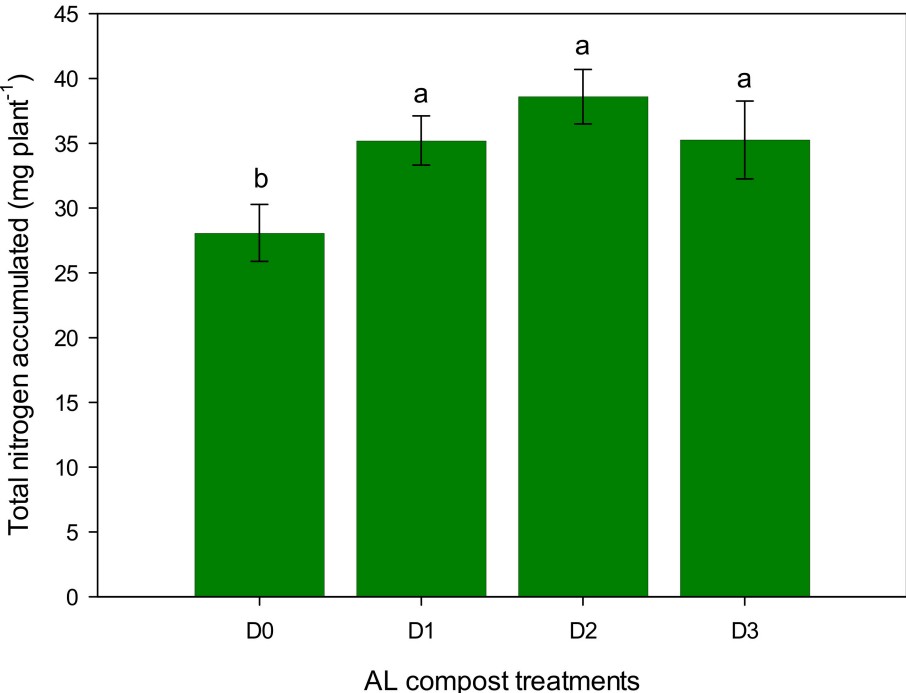

**Figure 3.** Total nitrogen accumulated ($T_{NA}$) in soybean plants grown with different doses of AL compost (D0, D1, D2 and D3). Similar letters are not statistically different according to the Tukey test ($p < 0.05$).

## 4. Discussion

Compost is an important source of organic matter, plant nutrients and plant growth-promoting microorganisms [13,14]. It has been demonstrated that compost is a valuable material for the production of commercial organic amendments and fertilizers [4,5], and also, an efficient alternative to peat as a carrier for rhizobia inoculants [15]. The AL compost used in our experiments presented an important organic matter content, but did not contain any rhizobia able to nodulate soybean plants (data not shown). However, as shown by the presence of nodules in the D1, D2 and D3 treatments, the inoculation of soybean with *B. diazoefficiens* helped to overcome the lack of the specific resident rhizobia in the AL compost where plants were grown. The absence of the proper rhizobia strain can be an obstacle for nodule formation, and so for the SNF. For this reason, recent studies have evaluated the improvement in nodule formation by the addition of the rhizobia strain able to nodulate the legume of interest. Ulzen et al. [8] demonstrated that soybean plants inoculated with *Bradyrhizobium* strain USDA 110 and grown in soil fertilized with fertisoil increased the number of nodules by 65% compared to non-inoculated plants. Moreover, the inoculation of *Medicago sativa* (cv. Siriver) plants with the *Rhizobium* strain (RhOL1) grown in soil amended with quack grass compost doubled the number of nodules compared to plants that were not inoculated [16]. These results highlight the requirement for the addition of inoculants containing the rhizobia strains to improve the effect of compost over nodulation efficiency.

In addition to the increment in nodule number, higher total nitrogen content was observed in treatments where compost was added (D1–D3). Similar results were found in faba bean grown in calcareous soil fertilized with traditional compost and vermicompost [17]. The soil residual nitrogen, and the nitrogen content in seeds and straws in plants fertilized with the two types of compost, were two to three times higher than in the control plants [17]. The effect of compost in plant growth has also been well documented. Compost promotes plant biomass, nutrient assimilation and crop yields. Specifically, compost improves legume crops such as common beans, soybeans, etc. [18,19]. Despite these advantages, compost presented a disadvantage related to nitrogen assimilation due to it being mostly under its organic form. Compost cannot be compared with mineral nitrogen fertilization during plant growth, but when both factors are combined (e.g., compost and nitrate), a synergetic effect is usually observed [14]. That means that the supplementation of compost with an easily-nitrogen form is required for relaying optimal crop yields, especially under greenhouse conditions. According to that, SNF could be a reliable strategy to achieve this goal.

Compost can improve SNF in soybeans through several mechanisms. Compost improves soil structure by increasing aggregation and promoting the formation of stable soil aggregates [20]. This allows for better aeration and water infiltration, which in turn can enhance the growth and activity of nitrogen-fixing bacteria, as well as the formation of root nodules. Compost increases the organic matter content of the soil, which can help to enhance the soil's water-holding capacity and provide a favorable environment for nitrogen-fixing bacteria [4]. Organic matter also serves as a food source for microorganisms, further supporting their growth and activity. Compost can suppress soilborne diseases and pests by promoting the growth of beneficial microorganisms that compete with or antagonize pathogens [13]. This can lead to healthier soybean plants with a better ability to establish symbiotic relationships with nitrogen-fixing bacteria. Finally, compost can stimulate root growth in soybean plants, increasing the surface area for nutrient uptake and nodule formation [8,15,16]. A more extensive root system can provide more opportunities for nitrogen-fixing bacteria to colonize and establish a symbiotic relationship with the plant.

Our data demonstrate, for the first time, that AL compost can improve soybean plant growth, as well as their nodulation capacity. Nitrogen accumulation was also increased by compost addition compared to the control treatment (without compost). As far as we know, the contribution of AL compost to the nitrogen fertilization of plants grown under greenhouse is scarce [14]. Therefore, we can conclude that most of the nitrogen of the soybeans derived from SNF, and that AL compost contributes to the improvement of SNF of the soybean-*B. diazoefficiens* symbiosis.

## 5. Conclusions

In this study, the effect of AL compost on the nodulation and SNF of soybean—*B. diazoeffciens* symbiosis was evaluated. The addition of AL compost enhanced the growth of soybean plants as it was revealed by the plant biomass augmentation and plant height. Furthermore, the nodule effectiveness and the symbiotic efficiency, measured by nodular leghemoglobin content and $T_{NA}$, respectively, were increased by AL compost. Despite this, the study was carried out under greenhouse conditions (microcosms), the results showed clear evidence of the synergetic effect of AL compost on soybean SNF. Further research needs to be carried out in order to confirm its viability under real conditions.

**Author Contributions:** Conceptualization, methodology, investigation, writing original draft, visualization, G.T.; Resources, writing review and editing, Funding acquisition, S.M. and M.J.D.; Conceptualization, writing review and editing, C.V.A.-G. All authors have read and agreed to the published version of the manuscript.

**Funding:** This work was funded by FEDER-co-financed grants AGL2017-85676-R and PID2021-1240070B to M.J.D. and grant PID2020-114330GB-100 to S.M. (Ministerio de Ciencia e Innovación [MICINN], Spain). Grants P18-RT-1401 to M.J.D., and S.M. from the Junta de Andalucía, Spain, and Tv22 to C.V.A.-G. funded by the Ministry of Agriculture and Rural Development of Colombia and Corporación Colombiana de Investigación Agropecuaria—Agrosavia, are also acknowledged. Analysis Service of the EEZ-CSIC thanks to grant EQC2019-005472-P funded by MCIN/AEI/10.13039/501100011033 and, as appropriate, by "ERDF A way of making Europe".

**Institutional Review Board Statement:** Not applicable.

**Informed Consent Statement:** Not applicable.

**Data Availability Statement:** Not applicable.

**Acknowledgments:** G.T. thanks to Paloma Pizarro-Tobias her comments and English improvements.

**Conflicts of Interest:** The authors declare no conflict of interest. The funders had no role in the design of the study; in the collection, analyses, or interpretation of data; in the writing of the manuscript; or in the decision to publish the results.

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
