# Peer review of "“Alperujo” Compost Improves Nodulation and Symbiotic Nitrogen Fixation of Soybean Inoculated with Bradyrhizobium diazoefficiens"

_nitrogen, doi:10.3390/nitrogen4020015_

Round 1
Reviewer 1 Report
The authors evaluated “Alperujo” Compost Improves Nodulation and Symbiotic Nitrogen Fixation of Soybean Inoculated with Bradyrhizobium diazoefficiens. There are many studied about compost and novelty of the paper is not clear.
In general, the information presented in the Introduction section is very general, and the research progress of the effect of compost on nitrogen fixation is not fully described. The sentences and the ideas are not well connected, not addressing well the hypothesis and objectives.
The growth period of soybean is only 28 days in this study. The growth time of soybean is too short to support the results of this paper. Therefore, I recommend it not to be published in Agronomy.
Specific points
1、 Reduce the research purpose and add the important results in abstract.
2、 The introduction should be re-writing to be more detail about the progress of compost and address the objectives. Knowledge gaps could be identified more clearly and then make more specific how your research aims to help close those knowledge gaps.
3、 How to measure leghaemoglobin concentration in L100 in Materials and Methods?
4、 Calculation formula of total nitrogen accumulated is not correct. It should include dry weight.L104-105
5、 What is the size of the pot used in this experiment? L110
6、 Some units are not acceptable. For instance ,unit of N content in Table3 is not correct. It should be g/kg.
7、 Discussion L188-182 are repeated results. Please simplify these sentences. Please strengthen the analysis of the reason why compost could improve nodule formation.
8、 The main findings in conclusion are missing.
Author Response
Review Report Form (reviewer 1)
Comments and Suggestions for Authors
The authors evaluated “Alperujo” Compost Improves Nodulation and Symbiotic Nitrogen Fixation of Soybean Inoculated with Bradyrhizobium diazoefficiens. There are many studied about compost and novelty of the paper is not clear.
Dear reviewer,
First of all, thanks for your comments and suggestions. In this “short communication”, we presented the results about how the nodulation and symbiotic nitrogen fixation (SBN) in the Soybean-Bradyrhizobium diazoefficiens symbiosis can be improved by using compost as an alternative to mineral fertilisers. It is true that there exist several studies about compost and legumes, but not about soybean and alperujo (AL) compost. This is the main novelty of the research. Also, the effect of AL compost on nodule fitness and functionality, assessed by nodular leghemoglobin, is quite relevant. As far as we know, our data demonstrate for the first time that AL compost can improve soybean plants growth, as well as their nodulation capacity.
In general, the information presented in the Introduction section is very general, and the research progress of the effect of compost on nitrogen fixation is not fully described. The sentences and the ideas are not well connected, not addressing well the hypothesis and objectives.
Thanks for your comments. This article in a “short communication”. This type of articles should include, in the Introduction, only the most relevant information about the studied topic. In a normal article, you can add as much information as you want or need. In our case, the topics that we comment are the relevance of legumes and the SNF (1st paragraph), the importance of AL compost (2nd paragraph) and the main effect of compost on SNF (3rd paragraph). The last paragraph is related to the objective of the research, which is the effect of AL compost in nodulation and SNF on soybean-B.diazoefficiens symbiosis.
The growth period of soybean is only 28 days in this study. The growth time of soybean is too short to support the results of this paper.
Unfortunately, we disagree with the reviewer. Our interest is focused on nodulation and nitrogen fixation process, which occurred during the first month of inoculation. This is demonstrated in our previous articles, such us Mesa et al. (2004). Physiol. Plant. 120, 205–211. https://doi.org/10.1111/j.0031-9317.2004.0211.x. Despites that, we have included this sentences in Conclusions as futures perspective:
Despite this study was carried out under greenhouse conditions (microcosms), the results showed a clear evidence of the synergetic effect of AL compost on soybean SNF. Further research needs to be done in order to confirm its viability under real conditions.
Therefore, I recommend it not to be published in Agronomy.
We are totally agreed with the reviewer. This is a “short communication” sent to Nitrogen journal and not to Agronomy journal.
Specific points
1- Reduce the research purpose and add the important results in abstract.
Thanks for the comment. We have eliminated this sentences:
At the end of the experiment, the dry weight and nitrogen content of plant biomass (aerial part and roots), the number and fresh weight of nodules, and the content of the nodule leghaemoglobin were determined.
And added this new one:
AL compost improved all the physiological parameters, as well as leghaemoglobin content (round 7 %) and the total nitrogen fixed (round 40 %).
2- The introduction should be re-writing to be more detail about the progress of compost and address the objectives. Knowledge gaps could be identified more clearly and then make more specific how your research aims to help close those knowledge gaps.
Thanks for your comments. This article in a “short communication”. This type of articles should include, in the Introduction, only the most relevant information about the topic studied. In a normal article, you can add as much information as you want or need. In our case, the topics that we comment are the relevance of legumes and the SNF (1st paragraph), the importance of AL compost (2nd paragraph) and the main effect of compost on SNF (3rd paragraph). The last paragraph is related to the objective of the research, which is the effect of AL compost in nodulation and SNF on soybean-B.diazoefficiens symbiosis.
3- How to measure leghaemoglobin concentration in L100 in Materials and Methods?
Thanks for the comment. The nodular leghaemoglobin determination is fully described in cite [12] Tortosa et al (2021). Nitrogen 2021, 2, 30–40, doi:10.3390/ni-265 trogen2010003. In our previous article, all the laboratory details are shown.
Despite that, we have included this information in the text:
by analyzing leghaemoglobin concentration (Lb) fluorimetrically after acidic reaction of the nodular fraction as it is described in [12]
4- Calculation formula of total nitrogen accumulated is not correct. It should include dry weight. L104-105
Thanks for the comment. All data of the article are referred to dry weight, as well as this formula. We have included this sentence to avoid misunderstanding:
Data were expressed as mg N plant-1 (dry weight).
5- What is the size of the pot used in this experiment? L110
Thanks for the comment. The volume of the pots was 0.25L. We have included this sentences in the Material and Methods:
Seeds were transplanted into sand-filled pots (0,25 L) and inoculated with 1 mL of B. diazoefficiens inoculum.
6- Some units are not acceptable. For instance, unit of N content in Table3 is not correct. It should be g/kg.
Thanks for the comment but we disagree with the referee. Table 3 shows N content in mg plant-1 due to these data are used for TNA determination (Figure 3). The N content expressed by g/kg can be easily calculate by dividing data of Table 3 by Table 1.
7- Discussion L188-182 are repeated results. Please simplify these sentences. Please strengthen the analysis of the reason why compost could improve nodule formation.
Thanks for the comment. We have included this paragraph about how compost can improve nodulation:
Compost can improve SNF in soybeans through several mechanisms. Compost improves soil structure by increasing aggregation and promoting the formation of stable soil aggregates [20]. This allows for better aeration and water infiltration, which in turn can enhance the growth and activity of nitrogen-fixing bacteria, as well as the formation of root nodules. Compost increases the organic matter content of the soil, which can help to enhance the soil's water-holding capacity and provide a favourable environment for nitrogen-fixing bacteria [4]. Organic matter also serves as a food source for microorganisms, further supporting their growth and activity. Compost can suppress soilborne diseases and pests by promoting the growth of beneficial microorganisms that compete with or antagonize pathogens [13]. This can lead to healthier soybean plants with a better ability to establish symbiotic relationships with nitrogen-fixing bacteria. Finally, compost can stimulate root growth in soybean plants, increasing the surface area for nutrient uptake and nodule formation [8,15,16]. A more extensive root system can provide more opportunities for nitrogen-fixing bacteria to colonize and establish a symbiotic relationship with the plant.
8- The main findings in conclusion are missing.
Thanks for the comment. We have rewritten the Conclusions like that:
In this study, the effect of AL compost on nodulation and SNF of soybean- B.diazoeffciens symbiosis has been evaluated. The addition of AL compost enhanced the growth of soybean plants as it was revealed by the plant biomass augmentation and plant height. Also, the nodule effectiveness and the symbiotic efficiency, measured by nodular legheaemoglobin content and TNA respectively, were increased by AL compost. Despite this study was carried out under greenhouse conditions (microcosms), the results showed a clear evidence of the synergetic effect of AL compost on soybean SNF. Further research needs to be done in order to confirm its viability under real conditions.
Reviewer 2 Report
General comment:
§ The work cannot be accepted in its current state. MS should be revised thoroughly before being submitted again for revision
Specific comment
§ Keywords as per journal format. Moreover 5 keywords are enough.
§ Mentioned greenhouse experimental pot condition viz., experimental design, amount of water regularly given, optimum humidity, light and dark hour per day, temperature, longitude and latitude positions of places where the experiments was performed, etc
§ Include the company names and the nation from where the equipments were purchased. Whenever applicable.
§ Include the reference of Dumas method
§ Mentioned the full name of abbreviation, whenever use first time.
§ Kindly include the value of standard deviation/error of each treatment in table.
§ Discussion need to the elaborate mechanism.
§ Conclusion are too short. It need to be elaborate. Moreover, the future prospective/ application should also need to be include in conclusion part.
Author Response
Review Report Form (reviewer 2)
Comments and Suggestions for Authors
General comment:
The work cannot be accepted in its current state. MS should be revised thoroughly before being submitted again for revision
Thanks for the comment
Specific comment
1- Keywords as per journal format. Moreover 5 keywords are enough.
Thanks for the comment. According to the Instructions for Authors section (https://www.mdpi.com/journal/nitrogen/instructions), we can use three to ten keywords. We prefer to keep our selection if it is possible.
2- Mentioned greenhouse experimental pot condition viz., experimental design, amount of water regularly given, optimum humidity, light and dark hour per day, temperature, longitude and latitude positions of places where the experiments were performed, etc.
Thanks for the suggestion. All this information can be found in 2.3 and 2.5 sections. We have also added the website of our Greenhouse and Growth Chamber Service of the Estación Experimental del Zaidín (EEZ-CSIC; https://www.eez.csic.es/en/green-houses) in case you need more information.
3- Include the company names and the nation from where the equipment were purchased. Whenever applicable.
Thanks for the comment. Most of the instruments used in this studied are cited. The only equipment that we can add the nation is LECO TruSpec CN Elemental Analyzer (USA)
4- Include the reference of Dumas method.
Thanks for the comment. We have added the [11] reference to this part.
5- Mentioned the full name of abbreviation, whenever use first time.
Thanks for the comment. We have added Estación Experimental del Zaidín (EEZ-CSIC) (lines 96-97) and analysis of variance (ANOVA) (lines 117-118).
6- Kindly include the value of standard deviation/error of each treatment in table.
Thanks for the suggestions, but we think this information is redundant with the statistical analysis. It is not common to show both. We prefer to keep our style if it is possible.
7- Discussion need to the elaborate mechanism.
Thanks for the comment. We have included this paragraph about how compost can improve nodulation:
Compost can improve SNF in soybeans through several mechanisms. Compost improves soil structure by increasing aggregation and promoting the formation of stable soil aggregates [20]. This allows for better aeration and water infiltration, which in turn can enhance the growth and activity of nitrogen-fixing bacteria, as well as the formation of root nodules. Compost increases the organic matter content of the soil, which can help to enhance the soil's water-holding capacity and provide a favourable environment for nitrogen-fixing bacteria [4]. Organic matter also serves as a food source for microorganisms, further supporting their growth and activity. Compost can suppress soilborne diseases and pests by promoting the growth of beneficial microorganisms that compete with or antagonize pathogens [13]. This can lead to healthier soybean plants with a better ability to establish symbiotic relationships with nitrogen-fixing bacteria. Finally, compost can stimulate root growth in soybean plants, increasing the surface area for nutrient uptake and nodule formation [8,15,16]. A more extensive root system can provide more opportunities for nitrogen-fixing bacteria to colonize and establish a symbiotic relationship with the plant.
8- Conclusion are too short. It need to be elaborate. Moreover, the future prospective/ application should also need to be include in conclusion part.
Thanks for the comment. We have rewritten the Conclusion like that:
In this study, the effect of AL compost on nodulation and SNF of soybean- B.diazoeffciens symbiosis has been evaluated. The addition of AL compost enhanced the growth of soybean plants as it was revealed by the plant biomass augmentation and plant height. Also, the nodule effectiveness and the symbiotic efficiency, measured by nodular legheaemoglobin content and TNA respectively, were increased by AL compost. Despite this study was carried out under greenhouse conditions (microcosms), the results showed a clear evidence of the synergetic effect of AL compost on soybean SNF. Further research needs to be done in order to confirm its viability under real conditions.
Round 2
Reviewer 1 Report
The quality of the manuscript has been improved since the previous version,yet further revisions are needed.
1 The abstract still needs to be modified. The main findings of the research and their importance are not clear. Line 22 two“.”. Delete one.
2 line 110-112, the unit of compost dosage should be changed to g/kg or kg/ha. Since we don’t know the size of the pot and how weight of soil used in each pot. Unit of g/pot is not clear。
Author Response
The quality of the manuscript has been improved since the previous version,yet further revisions are needed.
1 The abstract still needs to be modified. The main findings of the research and their importance are not clear. Line 22 two“.”. Delete one.
Thanks a lot for the comments. We have rephrased the abstract like that:
The utilization of compost to enhance plant productivity and symbiotic nitrogen fixation (SNF) is recognized as an effective alternative to synthetic nitrogen fertilizers. This environmentally sustainable method is readily accessible to farmers. This study investigated the effect of olive pomace ("alperujo", AL) compost on nodulation and SNF of soybeans (Glycine max L.) and their natural symbiont (Bradyrhizobium diazoefficiens). For that, soybean plants were subjected to several doses of AL compost under controlled greenhouse conditions. At the end of the experiment, the dry weight of plant biomass (aerial part and roots), the number and fresh weight of nodules, and nitrogen and leghaemoglobin content were analyzed. Application of AL compost significantly improved soybean growth as demonstrated by an increase in both plant biomass and height. Furthermore, nodular leghaemoglobin content (7%) and nitrogen content (approximately 40%) were found to be enhanced by the addition of AL compost, indicating an increase in nodule effectiveness and symbiotic efficiency, respectively. Our results provide clear evidence of the synergetic effect of AL compost on the soybean-B. diazoefficiens association, probably due to AL compost improved soybean roots development, and rhizospheric organic matter and nutrients assimilation by rhizobia.
2 line 110-112, the unit of compost dosage should be changed to g/kg or kg/ha. Since we don’t know the size of the pot and how weight of soil used in each pot. Unit of g/pot is not clear。
Thanks for the comment. The pots (0,25L) has a 10 cm of diameter, that means an area of 78,53 cm2 or 7,85 10-7 ha. Taking this data, the doses of AL composts were 28, 56, 84 t ha-1 for D1, D2 and D3, respectively.
We have added this information in 2.5 section:
Five treatments were tested according to AL compost doses applied: Non-inoculated soybeans with no compost added (Control), inoculated soybean with B. diazoefficiens without compost (D0), inoculated soybean with B. diazoefficiens with compost (2.2 g per pot or 28 t ha-1) (D1), inoculated soybean with B. diazoefficiens with compost (4.4 g per pot or 56 t ha-1) (D2), and inoculated soybean with B. diazoefficiens with compost (6.6 g per pot or 84 t ha-1) (D3).

Reviewer 2 Report
The MS can be accepted in its present form.
Author Response
Thank you very much for all your corrections and suggestions.